# Comparison of Emotion Recognition in Young People, Healthy Older Adults, and Patients with Mild Cognitive Impairment

**DOI:** 10.3390/ijerph191912757

**Published:** 2022-10-05

**Authors:** Giulia Francesca Barbieri, Elena Real, Jessica Lopez, José Manuel García-Justicia, Encarnación Satorres, Juan C. Meléndez

**Affiliations:** 1Department of Nervous System and Behavioral Sciences, University of Pavia (Italy), Piazza Botta 6, 27100 Pavia, Italy; 2Department of Developmental Psychology, Faculty of Psychology, University of Valencia (Spain), Av. Blasco Ibañez 21, 46010 Valencia, Spain

**Keywords:** basic discrete emotions, facial emotion recognition, healthy older adults, mild cognitive impairment, young people

## Abstract

Background: The basic discrete emotions, namely, happiness, disgust, anger, fear, surprise, and sadness, are present across different cultures and societies. Facial emotion recognition is crucial in social interactions, but normal and pathological aging seem to affect this ability. The present research aims to identify the differences in the capacity for recognition of the six basic discrete emotions between young and older healthy controls (HOC) and mildly cognitively impaired patients (MCI). Method: The sample (*N* = 107) consisted of 47 young adults, 27 healthy older adults, and 33 MCI patients. Several neuropsychological scales were administered to assess the cognitive state of the participants, followed by the emotional labeling task on the Ekman 60 Faces test. Results: The MANOVA analysis was significant and revealed the presence of differences in the emotion recognition abilities of the groups. Compared to HOC, the MCI group obtained a significantly lower number of hits on fear, anger, disgust, sadness, and surprise. The happiness emotion recognition rate did not differ significantly among the three groups. Surprisingly, young people and HOC did not show significant differences. Conclusions: Our results demonstrated that MCI was associated with facial emotion recognition impairment, whereas normal aging did not seem to affect this ability.

## 1. Introduction

Emotions represent adaptation strategies in the species that use them. According to evolutionary theories, they evolved to promote survivability. The basic discrete emotions appear to be similar and comparable across different cultures and societies, and they are divided into six categories: happiness, disgust, anger, fear, surprise, and sadness [1]. Currently, several authors agree on defining them as pre-organized, involuntary, and rapid responses to stimuli that can positively or negatively affect the well-being of an organism [1,2]. The ability to recognize facial expressions of emotions in other people is a crucial component of interpersonal communication in humans, and, as shown in important literature, it is hindered by both normal and pathological aging processes [3,4]. Particularly, facial expressions can influence a person’s emotional experience by giving signals to others about how he/she feels.

Traditionally, this ability has been studied through easily controllable paradigms, such as Ekman’s basic facial emotion expression paradigm [5,6]. Most of these studies have shown age differences in facial emotion recognition capacity when identifying emotions such as anger, sadness, and fear [7], whereas recognition of happiness and disgust are preserved [5]. Accordingly, a recent study with a large sample size showed the presence of age differences in a traditional facial emotion recognition paradigm, with higher scores in the young adult group and progressively lower hits in middle-aged and older-adult groups [8]. Thus, according to more recent research, the ability to identify emotions peaks in young people between the ages of 15 and 30, and then progressively declines after the age of 30 [9].

Several authors have argued that changes in the ability to identify facial expressions of emotion could be related to age-related structural decay in neural systems in temporal, limbic, and prefrontal areas [10,11,12]. However, the positivity processing bias, that is, adults’ tendency to direct their attention to and memorize positive emotional information more than negative information, could also play a role [13,14]. Some studies suggest that age-related cognitive decline in processing speed and working memory could also ex-plain age differences, particularly with emotions that are more difficult to identify, such as disgust [3,15]. In line with this, some authors claim that distinct patterns of emotional impairment as a consequence of brain damage, for example to the amygdala, is related to impairment in processing specific emotional stimuli [16].

Currently, a growing body of studies is investigating the ways cognitive impairment can affect the processing of emotional stimuli [17]. It is well known that rare forms of dementia, such as frontal–frontotemporal, can alter social and emotional functioning. Moreover, a decrease in facial emotion processing ability has been found in psychiatric conditions, revealing a negative bias towards positive emotions in patients with depression and a more general impairment in emotion recognition in schizophrenia [18,19]. Mounting evidence has also demonstrated the presence of alterations in emotion processing in Alzheimer’s disease (AD) [20]. Studies have shown that, in emotional labeling tasks, AD subjects’ emotion perception ability is impaired, particularly when labeling faces that show sadness, anger, fear, and surprise [21]. In fact, a recent review [22] reported that, above all, the recognition of happiness remained the best preserved in the AD participants, whereas negative emotions appeared to be more difficult to identify.

Several studies have shown differences in facial emotion recognition abilities between mild cognitive impairment (MCI), early- and moderate-stage Alzheimer’s dementia (AD), and healthy older individuals. According to a review, MCI subjects have poorer emotion recognition performance than in normal aging, especially in detecting negative emotions [17]. In other studies, MCI patients showed poorer performance in recognizing low-intensity fear in faces and in labeling faces that showed fear, sadness, and anger, compared to healthy older controls [23,24]. Compared to healthy older controls, early-stage AD patients reported greater issues with labeling fearful, angry, and happy expressions [25]. In patients with mild to moderate Alzheimer’s, the facial emotion recognition deficits extended to sadness, surprise, and disgust [26].

The main structures involved in emotion processing seem to be the amygdala and orbitofrontal circuits [27], the insula and striatum [28], and the ventromedial prefrontal cortex [29]. The amygdala seems to be mainly involved in the recognition of expressions of fear, whereas the insula and basal ganglia [30,31] play a role in the identification of disgust. Sadness has been associated with activity in the anterior cingulate and subcallosal cingulate [32]. Impaired emotion recognition ability in neuropsychiatric disorders, such as schizophrenia and depression, has been attributed to alterations in brain activity in frontotemporal regions [33], whereas normal aging is associated with a decline in the orbitofrontal cortex, cingulate cortex, and amygdala, leading to difficulties with anger, sadness, and fear recognition, respectively [3]. In the first stages of the illness, Alzheimer’s disease damages the medial temporal lobe structures [6]. Some studies have found neurofibrillary tangles in the entorhinal cortex, hippocampus, and amygdala [34,35,36,37]. Thus, it has been hypothesized that brain decay observed in emotion processing structures is the basis for the decline in emotion recognition.

Impairments in emotion perception have been associated with interpersonal difficulties [38,39,40]. The integrity of the ability to correctly identify emotional stimuli is essential to social behavior competence. Damage to this ability is associated with depression, inappropriate social behavior, relationship problems, and psycho-behavioral disorders [38,40,41,42,43]. Social ability disorders are also associated with caregiver burden and a negative impact on quality of life [41,44]. In individuals affected by AD, the decline in social function leads to problematic interpersonal behaviors, such as anxiety, phobia, agitation, and aggressive behavior [41]. It is reasonable that impairments in emotion processing, especially in recognizing the emotional state of the other, involve relational problems with social functioning [40].

The purpose of this study is to compare the facial emotion recognition abilities in young healthy subjects, healthy older adults, and MCI patients using Ekman’s basic facial emotion expression paradigm [5]. In particular, the present research is designed to identify the differences in the recognition of the six basic discrete emotions. Based on previous evidence [3,4,7,15], we hypothesized that poorer performance in the recognition of anger, sadness, and fear would be found in healthy older adults compared to younger participants, whereas the level of recognition of disgust and happiness would be similar. Furthermore, in line with the literature, we expected to find lower recognition accuracy for the six basic discrete emotions in cognitively impaired subjects compared to healthy controls [3]. The present research aims to contribute evidence about how facial expression recognition changes during normative and initial pathological aging, providing useful information about possible changes that occur with age and cognitive decline.

## 2. Materials and Methods

### 2.1. Participants

The sample was composed of 107 participants: 47 young adults (19 men, 28 women), 27 healthy older adults (12 men, 15 women), and 33 mildly cognitively impaired patients (17 men, 16 women). The young adults, with ages ranging between 18 and 26 years (*M*_age_ = 19.59, *SD* = 2.2), were psychology students from the University of Valencia. The healthy older adults (HOA), with ages ranging between 60 and 87 years (*M*_age_ = 67.67, *SD* = 6.5), were students from the Nau Gran (a free university academic course for people over 55 years old) of the University of Valencia. Mildly cognitively impaired (MCI) patients, with ages ranging between 64 and 88 years (*M*_age_ = 74, *SD* = 6.9), were from the Neurology Department of the General Hospital of Valencia.

The inclusion criteria for young adults were: age between 18–30 and score on the Mini Mental State Examination (MMSE) [45] > 23 (*M*_Young_ = 29.8, *SD* = 0.5, range: 28–30); for healthy older adults, the inclusion criteria were: age > 60, score on the MMSE > 23 (*M*_HOA_ = 28.88, *SD* = 1.2, range: 26–30), and levels 1 or 2 on the Global Deterioration Scale (GDS) [46]. Patients in the MCI group had to meet the following criteria: age > 60, score on the MMSE between 26–18 (*M*_MCI_ = 23.25, *SD* = 3.1, range: 18–26), and level 3 on the GDS. A clinical diagnosis was also the end result of an extensive evaluation, including medical history and physical and neuropsychological examinations, and it was determined by consensus between neurologists and a neuropsychologist. Exclusion criteria for patients were: significant asymptomatic neurovascular disease, a history of previous symptomatic stroke, any medical condition significantly affecting the brain, or serious psychiatric symptoms. All the participants were informed about the study protocols and signed an informed consent form to take part in the study. All the procedures followed the tenets of the Declaration of Helsinki.

### 2.2. Materials

#### 2.2.1. Cognitive Measures

First, a complete cognitive assessment of the participants was carried out. Except for the Global Deterioration Scale [46], which was only administered to the HOA and MCI groups, the following neuropsychological scales were included in the battery designed to assess the cognitive state of the participants.

The Global Deterioration Scale [46] assesses the severity of primary degenerative dementia and delineates seven clinically distinguishable global stages ranging from normality (1) to severe dementia of the Alzheimer (AD) type (7). The Global Deterioration Scale analyzes patients’ ability to function, reflected in daily living and instrumental activities, as well as psychiatric morbidity based on progressive cognitive loss.

The Mini Mental State Examination (MMSE) [45] was administered to obtain an index of overall cognitive functioning. The MMSE is a screening test that quantitatively estimates the existence and severity of cognitive impairment. The maximum score is 30 points, which is obtained by adding up the scores on all the items. The cut-off score for cognitive impairment is usually set at 23 points.

The Memory Alteration Test (M@T) [47] consists of five blocks of testing: immediate memory, temporal oriented memory, semantic remote memory, free recall memory, and recall memory with cues. The M@T is a cognitive screening measure with high discriminant value for amnesic-type mild cognitive impairment and mild Alzheimer’s disease in the general population. The total score is 50, and to distinguish amnestic-type mild cognitive impairment from subjective memory complaints, the optimal cut-off is 37 points. The test makes it possible to obtain partial scores for each block.

The Spain-Complutense Verbal Learning Test (TAVEC) [48] consists of subjects learning a list of words, which are presented as a “shopping list” read several times by the examiner. The test begins by reading aloud a list of 16 words that the participant is asked to repeat; this procedure is repeated five times. Twenty minutes after the immediate recall and learning phase, the participant is asked to say the words he/she remembers, and delayed recall is assessed.

The Wechsler Adult Intelligence Scale-III (WAIS III) [49] digits subtest makes it possible to evaluate attentional capacity and immediate and working memory skills by presenting the subject with gradually increasing amounts of information. On the direct digits task, the subject must repeat the sequence of numbers in the same order in which it is read by the examiner. On the inverse digits task, the subject must say the numbers in the inverse order to that read by the examiner, and so greater working memory processing is required. On both parts, the test has eight elements containing two items each; one point is given for each correct item, with a maximum score of 16 on both. These would be percentile scores, which are later converted to standard scores using the table of scales.

The Rey Figure Test [50] measures memory and other executive functions. The subject must carefully reproduce a complex geometric drawing and later (3 min) reproduce it from memory. The first part evaluates cognitive aspects, such as planning, motor skills, operative memory, and visuo-constructive and spatial skills. The total of all the scores assigned to all the elements yields the percentile score, which is then converted into a standard score where 50 is the cut-off point.

Barcelona Test Revised [51]. The subtests of categorical fluency and phonological fluency were used. To measure categorical fluency, the subject has to evoke the greatest number of words linked to a specific category within a limited period of time. In this study, the subject is asked to name animals for one minute. In the case of phonological fluency, the subject is asked to name words beginning with the letter “p” for a maximum of three minutes. Thus, it is possible to observe the capacity for accessing and evoking elements from the lexical and semantic warehouse.

#### 2.2.2. Emotion Assessment

The participants’ Facial Emotion Recognition ability was successively assessed by administering the Ekman 60 Faces test [52]. The task consists of a set of 60 photos of 10 human faces (4 men, 6 women) depicting the six basic emotions (happiness, sadness, fear, surprise, anger, and disgust), and participants have to identify the emotion of the expression in the picture by choosing from six labels. The pictures are based on a Series of Pictures of Facial affect, a widely used and validated series of photographs in facial expression research [5]. In the present study, on a computer screen, participants were shown the pictures for 5 s in a randomized order, following the original procedure [5] Then, with the keyboard, from the six labels representing the Spanish word for each basic discrete emotion, participants had to choose the one that best suited the emotion depicted in the picture. A selection had to be made before proceeding to the following picture. An example trial was planned before the actual testing phase. It consisted of six sample trials showing a facial expression for each basic discrete emotion, but using a face that was not included in the testing phase. All the answers on the example trials were excluded from the analysis. The total score on the test is 60, that is, the best possible performance, and each emotion has a sub-score of 10 points.

### 2.3. Analyses

For the study, two MANOVAs were performed to determine whether there were differences between groups on the cognitive variables and emotion recognition. Subsequently, ANOVAs were conducted to find out which dimensions showed differences, and post-hoc Bonferroni tests were performed to find the differences between the groups in the variables that had shown differences in the ANOVA.

## 3. Results

### 3.1. Cognitive Measures Results

First, a MANOVA was conducted to study the differences between the groups on the cognitive measures; the multivariate contrasts revealed a main effect of group (Λ = 0.066; *F*(20, 190) = 21.07; *p* < 0.001; η^2^ = 0.743). Therefore, ANOVAS were performed for each dependent variable, and significant differences were obtained on all the cognitive measures analyzed (Table 1).

Post hoc Bonferroni tests on the MMSE and M@T showed that young and healthy older adults did not differ from each other, but they scored significantly higher than the MCI group on both the MMSE (*p* < 0.001) and M@T (*p* < 0.001).

TAVEC scores for immediate memory showed differences between the young and healthy older adult groups (*p* = 0.022) and MCI (*p* < 0.001) and between healthy older adults and MCI (*p* < 0.001). On the TAVEC total word and delayed recall scores, young people scored significantly higher (*p* < 0.001) than healthy older adults and MCI, and there were also significant differences between healthy older adults and MCI (*p* < 0.001).

Young and healthy older adults did not differ on the Rey figure, but they scored significantly higher than MCI (*p* < 0.001); however, on the delayed test, young adults scored higher than healthy older adults and MCI (*p* < 0.001), and healthy older adults scored higher than MCIs (*p* < 0.001).

On direct and inverse digits, the MCI showed poorer scores than young people and healthy older people (*p* < 0.001), but no differences were observed between young and healthy older adults.

Finally, young people showed higher scores in both categorical fluency and phonological fluency than the healthy older adults (*p* = 0.017; *p* = 0.001) and MCI (*p* = 0.001; *p* < 0.001), but healthy older adults’ scores did not differ from MCIs’ scores.

### 3.2. Emotion Assessment Results

Next, a MANOVA was performed to compare the groups on emotion recognition, with significant differences found for the group variable (*F*(12, 200) = 7.96; *p* < 0.001; η^2^ = 0.323). Subsequently, the inter-subject effects were analyzed and showed differences between the groups (Figure 1) in recognizing anger (*F*(2, 104) = 18.77; *p* < 0.001; η^2^ = 0.265), fear (*F*(2, 104) = 25.54; *p* < 0.001; η^2^ = 0.329), surprise (*F*(2, 104) = 14.71; *p* <.001; η^2^ = 0.221), disgust (*F*(2, 104) = 17.77; *p* <.001; η^2^ = 0.255), and sadness (*F*(2, 104) = 28.52, *p* < 0.001, η^2^ = 0.354), whereas they did not show significant differences in recognizing happiness (*F*(2, 104) = 2.43; *p* =.092; η^2^ = 0.045).

Post hoc Bonferroni tests were performed, and significant differences in anger recognition were found between young adults and MCI (*p* < 0.001) and between healthy older adults and MCI (*p* < 0.001). In this case, the accuracy means of the young people (*M* = 8.21, *SD* = 1.2) and healthy older adults (*M* = 8.22, *SD* = 1.4) were higher than those of MCI patients (*M* = 6.00, *SD* = 2.4).

In addition, with fear recognition, post-hoc comparisons revealed significant differences between the young people and the MCI group (*p* < 0.001) and between healthy older adults and MCI (*p* < 0.001). The MCI patients’ mean accuracy on fear recognition was lower (*M* = 4.21, *SD* = 2.5) than that of the young people (*M* = 7.57, *SD* = 1.6) and healthy older adults (*M* = 6.56, *SD* = 2.2).

Similarly, in the case of surprise, differences were found between young adults and MCI (*p* < 0.001) and between healthy older adults and MCI (*p* = 0.002). The means for both the young people (*M* = 9.23, *SD* = 1.1) and the healthy older adults (*M* = 8.74, *SD* = 1.1) were higher than the MCI group’s mean (*M* = 7.36, *SD* = 2.3).

Regarding disgust, the Bonferroni analysis revealed that young and healthy older adults differed significantly from the MCI patients (*p* < 0.001); both had higher means (*M* = 7.45, *SD* = 1.8, *M* = 8.30, and *SD* = 1.4, respectively) than the patient group (*M* = 5.15 and *SD* = 3).

Finally, in sadness recognition, significant differences were found, showing that both young and healthy older adults differed from MCI (*p* < 0.001). The MCI patients’ mean (*M* = 5.55, *SD* = 2.1) was lower than the means of the young adults (*M* = 8.23, *SD* = 1.5) and healthy older adults (*M* = 8.11, *SD* = 1.4).

## 4. Discussion

In the present research, we compared the facial emotion recognition ability of young people, healthy older adults, and MCI patients using Ekman’s 60 faces paradigm. Specifically, the main objective of the research was to identify how the ability to recognize facial expressions of emotions changes in normal and pathological aging. Knowledge about the way emotion perception changes across the life cycle can help clinicians and caregivers to develop targeted interventions and new compensatory communication.

Consequently, based on previous evidence, we hypothesized that we would find differences between the three groups, and in particular, weaker performance in the group of MCI patients compared to young people and healthy older participants. Moreover, we expected to find weaker performance in recognition of anger, sadness, and fear in healthy older adults than in younger participants, whereas their levels of disgust and happiness recognition were expected to be similar.

Supporting our main hypothesis, the results of the MANOVA analysis were significant and revealed the presence of differences in emotion recognition ability among the groups. The ANOVA tests revealed that the rate of recognition of the happiness emotion did not differ significantly among the three groups. Regarding the other five basic discrete emotions, post-hoc analysis showed no differences between young people and healthy older adults. However, compared to healthy controls, MCI patients obtained a significantly lower number of hits. Surprisingly, the facial emotion recognition performance of young participants and healthy older adults did not show any significant differences; they had similar rates of hits on all six basic discrete emotions.

In general, our results are consistent with the reviewed literature, which reports the presence of changes in facial emotion recognition ability with both normal and pathological aging. Specifically, our data are in line with current evidence showing that, from the first stages of the illness, Alzheimer’s disease compromises the ability to correctly identify facial expressions of emotion [17,22]. For example, previous research showed that, compared to healthy older controls, MCI patients’ facial emotion recognition was impaired, particularly for negative emotions [23,24]. Similarly, we showed that compared to healthy controls, our group of patients had trouble identifying the basic discrete emotions, namely, anger, fear, surprise, disgust, and sadness, but not happiness. Consistent with our results, a recently published review [22] postulated that AD patients’ ability to recognize emotion is impaired, especially when labeling negative emotions, whereas the ability to correctly identify happiness remains the most intact.

However, there are still conflicting results with two main explanatory hypotheses. Some authors suggest that difficulties with emotional recognition are due to problems associated with the cognitive impairment itself, whereas others attribute them to neurodegenerative changes in the emotional processing capacity [6,26]. On the one hand, studies have attributed observed decreases in emotion recognition to general cognitive, linguistic, or visuospatial functions rather than to specific emotional processing deficits, observing, for example, correlations between cognitive measures and impaired emotion recognition ability [17]. On the other hand, scientific evidence shows that regions of the brain responsible for the identification of emotions may be impaired to some degree. For example, it has been demonstrated that Alzheimer’s Disease, even from the very early phases, targets medial temporal lobe structures. Because these areas are directly involved in emotion recognition, this skill may be diminished. Moreover, evidence from studies on schizophrenia and depression has linked dysfunction in frontal brain activity to facial emotion recognition impairment, supporting this hypothesis [33].

Furthermore, it can be argued that patients’ impairment in emotion recognition could be related to the kind of stimuli employed. For example, a recent study [53] showed that, compared to controls, AD patients’ ability on emotional Stroop tasks tends to be impaired, especially with facial stimuli, whereas with words as stimuli, they showed less interference in identifying conflicting stimuli. In this regard, the authors concluded that an emotion recognition deficit could be less evident when the communication is through words, and not merely through facial expressions. Moreover, in patients with AD, studies using voice and music stimuli have shown a preserved emotional recognition ability [54,55]. Consequently, it can be argued that AD patients’ recognition ability through dynamic auditory emotions may be more preserved than with static visual emotions [54]. In this regard, evidence from a recent review supports the beneficial effect of dynamic information on recognition accuracy, showing greater coherence in healthy participants’ performance when identifying specific emotions [56]. These findings shed light on the need to conduct further studies with more ecologically valid stimuli in people with AD and MCI, and in healthy people.

Investigating the influence of characteristics of emotional stimuli on mildly cognitively impaired patients—and, of course, on elderly people—could help to clarify how to interpret the results in future studies.

Unexpectedly, our results were not consistent with previous evidence showing age-related differences in facial emotion recognition. Our study showed similar performances in young and older participants with all six basic discrete emotions, including anger, surprise, and fear. It should be noted that the healthy older adult sample enrolled in the present research was involved in a large number of cognitive leisure activities because they participated in free courses at the university. Thus, the participants’ preserved emotion recognition abilities could be attributed to Stern’s cognitive reserve concept, which explains individual differences in the susceptibility to age-related changes and brain pathologies, such as those observed in Alzheimer’s disease [57]. According to Stern’s cognitive reserve hypothesis, factors such as the level of education, employment reached, and cognitive leisure activities are able to increase reserve and allow individuals to tolerate higher levels of brain decay, keeping cognitive functioning intact. Therefore, it can be argued that our sample of healthy older subjects, who were involved in modern, up-to date activities, could have an advantage in traditional emotional face recognition paradigms, thus representing a novelty in the study of aging’s influence on facial emotional recognition.

However, the present study supports the evidence about the preservation of the ability to recognize disgust and happiness. Our data are consistent with research that points out that age-related brain structure changes do not affect the insula in the same way that they affect other brain areas, resulting in a lasting ability to identify disgust [29]. Moreover, there is similarity with research showing that, in contrast to negative emotion recognition, age-related changes are not observed in happiness recognition [58], although the underlying reason is still being debated. For example, some research took into account that older adults tend to be more oriented towards positive stimuli than towards negative, resulting in better performance in happiness recognition than in anger, sadness, and fear recognition [14].

In concrete terms, our results suggest that impairments in emotion recognition could be more prominent in the clinical population. Specifically, the present study adds further evidence to the growing research on deficits in facial emotion recognition due to cognitive impairment in clinical populations. Furthermore, it contributes novel evidence of interest to the existing literature on age-related changes in emotion recognition, showing the possibility of maintaining this ability in healthy aging. Thus, our findings imply, to a certain extent, that emotion recognition disorders could signal the presence of mild cognitive impairment, even in the early stages. In this regard, the identification of impairments in emotional recognition ability could be important in the early diagnosis of cognitive decline, promoting the individual’s early involvement in appropriate interventions in order to alleviate the impact of cognitive impairment on society and individuals.

As the evidence suggests, despite their lower performance in facial emotion recognition tasks, healthy older adults do not experience social interaction difficulties in everyday life [8]. However, as previous research reported, MCI patients could be more likely to experience relationship problems due to difficulties with facial emotion recognition [39]. Whereas healthy older adults have less potential for conflict and higher levels of satisfaction in intimate relationships [8], it has been shown that, in more advanced stages of the illness, AD patients’ level of psychological well-being decreases, showing, above all, that there is a decline in the positive relationship dimension, as well as in autonomy, personal growth, and purpose in life [59]. In light of our results, it is important to be able to involve this category of patients, even at an early stage, in clinical interventions. Loneliness has serious consequences for emotion, behavior, morbidity, and cognition, and it has been associated with cognitive impairment, accelerated cognitive decline, and an elevated risk of Alzheimer’s disease. In this regard, the research recommends social integration because of its beneficial effects on health [59]). In this regard, reminiscence therapy is a suitable intervention because it offers the opportunity, through autobiographical reminiscence, to stimulate emotional processing skills, fostering social interaction and emotional exchange [60]. Specifically, integrative reminiscence intervention programs have been developed to help both clinical and normative populations. As a result, participants have shown great improvements in psychological, cognitive, and emotional wellbeing.

Our findings about MCI patients’ lower emotion recognition ability encourage the development of new research lines. Specifically, it would be interesting to relate our results to the study of changes in socio-cognitive mechanisms, such as the theory of mind, which can affect mildly cognitive impaired patients. Therefore, for further research, we recommend comparing MCI patients and populations with diminished or heightened theory of mind, such as people with autism and schizophrenia.

## 5. Conclusions

Mild cognitive impairment seemed to cause facial emotion recognition impairments in patients, whereas healthy aging was not found to have any particular effect on this ability. Given our results, future research should consider the role of cognitive reserve in decay patterns, encouraging cognitive reserve enhancement programs to prevent emotion recognition impairment and its effects on the individual’s well-being. In sum, knowledge about the patterns of change in the recognition of facial expressions in pathological and normal and healthy aging can help clinicians and caregivers to develop adequate interventions for its preservation and identify compensatory styles of communication.

## Figures and Tables

**Figure 1 ijerph-19-12757-f001:**
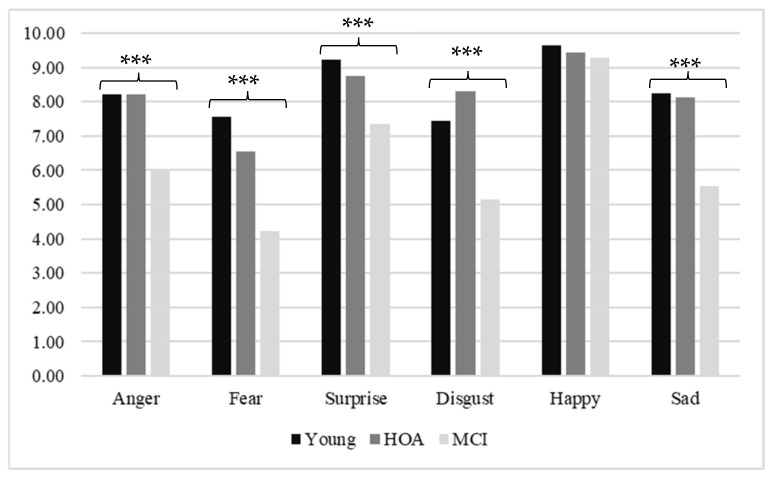
Means of emotions by group. The graph displays inter-subject effects, which revealed differences between groups in recognition of anger *** (*p* < 0.001), fear *** (*p* < 0.001), surprise *** (*p* < 0.001), disgust *** (*p* < 0.001), and sad *** (*p* < 0.001) emotions.

**Table 1 ijerph-19-12757-t001:** Descriptive statistics and results of the ANOVAs of the cognitive measures.

	Young	HOA	MCI	g.l.	F	*p*	η^2^
MMSE	29.74	28.88	23.25	2, 104	102.91	<0.001	0.696
M@T	47.51	45.50	26.72	2, 104	147.61	<0.001	0.766
TAVEC immediate	7.71	6.46	3.47	2, 104	50.30	<0.001	0.528
TAVEC total	61.71	52.08	23.81	2, 104	192.29	<0.001	0.810
TAVEC delayed	14.34	11.81	2.13	2, 104	231.18	<0.001	0.837
Rey’s Copy	35.60	34.76	28.21	2, 104	18.60	<0.001	0.292
Rey’s Delayed	28.92	22.67	5.85	2, 104	141.43	<0.001	0.759
Direct digits	10.14	10.46	7.16	2, 104	22.68	<0.001	0.335
Inverse digits	8.97	8.38	4.31	2, 104	37.03	<0.001	0.451
Categorical fluency	18.23	15.04	14.09	2, 104	8.30	<0.001	0.156
Phonological fluency	34.83	25.23	20.28	2, 104	17.96	<0.001	0.285

## Data Availability

The data presented in this study are available on request to the authors. The data are not publicly available due to privacy reasons.

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
