# Peer review of "Comparison of Emotion Recognition in Young People, Healthy Older Adults, and Patients with Mild Cognitive Impairment"

_ijerph, 2022, doi:10.3390/ijerph191912757_

Round 1
Reviewer 1 Report
The present study investigated the influence of age and mild cognitive impairment on emotion recognition, the results showed that individual’s emotion recognition performance was affected by mild cognitive impairment (but not age), this study is interesting, but the following issues need to be resolved.
1. The authors mentioned that “The ability to recognize facial expressions of emotions in other people is a crucial component of interpersonal communication in humans, and it is hindered by both normal and pathological aging processes” (lines 35-37), you should provide corresponding references.
2. The authors mentioned that “Moreover, two reviews reported that AD patients’ emotion labeling ability is also hindered by the intensity of the emotion displayed. Specifically, they showed that lower emotional intensity stimuli are associated with lower facial emotion recognition in AD patients [18, 19].”, I suggest deleting this sentence, considering that this point is not sufficiently relevant to this study.
3. When introducing the research related to impairments in emotion perception, the authors listed a large number of AD-related studies, I suggest you added the findings of related clinical findings such as depression and schizophrenia.
4. The authors mentioned that “It is plausible that impairments in emotion processing” (line 96), I don’t think this phenomenon is plausible, do you want to say this phenomenon is “reasonable”?
5. When the authors put forward their hypotheses, the corresponding basis should be clearly provided, that is, the corresponding references should be clearly given.
6. The sentence “Knowledge about the way emotion perception changes during the life cycle can be helpful to clinicians and caregivers in developing targeted interventions and new compensatory communication.” (line 109-111) should be moved to the Discussion section.
7. How did the authors determine the sample size? In particular, the sample sizes of different groups vary greatly. How do we know if the sample size is large enough? Whether participants were paid for their participation?
8. In Fig.1, the significant difference between different conditions should be marked by asterisk(s), i.e., “*”, “**”, or “***”.
9. The innovation and significance of this study should be clearly explained in the Discussion section.
10. Inappropriate use of punctuation. “Method.”, “Results.”, and “Conclusions.” should be replaced by “Method:”, “Results:”, and “Conclusions:”, respectively.
Grammatical errors. Such as “Our results demonstrate that…” (line 23), “our results suggest that …” (line 339), the correct saying are “demonstrated” “suggested”.
Reviewer 2 Report
This paper sought to assess the effects of normal and pathological aging on the recognition of emotions. It has important implications for individuals suffering from an ability (total or partial) to recognize emotions and adapt their behaviors thereto. Thank you for your important work.
In general:
The paper is very straight forward, well-written, and to the point. The message is extremely important, with both clinical and non-clinical implications.
The language, however, could be slightly improved by editing from a native English speaker. For example, “the ability to correctly identify happiness is spared the most” would read much better as “the ability to correctly identify happiness remains the best preserved among participants”.
Introduction:
In the introduction, you mention neuroanatomical correlates of emotion recognition. Might you make some space for a quick discussion of genetic correlates as well? There is much research in this space that could serve to be mentioned – especially because the complex (epi)genetics are what primarily underpin the neuroanatomical differences.
Methods and Results:
These are sound and well-presented.
Discussion:
While your research focuses on the recognition of facially expressed emotions, it would be interesting to assess emotions expressed via other sensory modalities. For example, do the deficits apply to the recognition of an emotionally salient tone of voice, pitch, or rhythm? What about an emotionally charged kinesthetic expression (e.g. action, movement)? Or touch? These would be fascinating to probe deeper given the implications for 1) how we interpret emotions and 2) how we can help train individuals with a deficit in one to make up for it by paying closer attention to cues in the other modality.
A few broad sentences on the implications of your findings beyond aging populations would be interesting. For example, how does this research apply to populations with diminished theory of mind (e.g. autism) or heightened theory of mind (e.g. schizophrenia)?
(More broadly, how might your findings relate to the results from Baron-Cohen’s “reading the mind in the eyes” test? Such cross-disciplinary comparisons would highly enrich your discussion.)
Can your research pave the way for training programs to boost emotion recognition across all clinical and non-clinical populations, regardless of age, diagnosis, gender?
Specifically as regards an aging population, would cognitive training be expected to circumvent the losses in emotion recognition resulting from cognitive decline? This would flow well as a detailed dicussion following your comment, “For this reason, the early diagnosis of emotion recognition disturbances could help clinicians to implement appropriate interventions to ease the impact on society and individuals.”
